# Effect of Humic Acid on Soil Physical and Chemical Properties, Microbial Community Structure, and Metabolites of Decline Diseased Bayberry

**DOI:** 10.3390/ijms232314707

**Published:** 2022-11-25

**Authors:** Haiying Ren, Mohammad Shafiqul Islam, Hongyan Wang, Hao Guo, Zhenshuo Wang, Xingjiang Qi, Shuwen Zhang, Junning Guo, Qi Wang, Bin Li

**Affiliations:** 1Institute of Horticulture, Zhejiang Academy of Agricultural Sciences, Hangzhou 310021, China; 2State Key Laboratory for Managing Biotic and Chemical Treats to the Quality and Safety of Agro-Products, Hangzhou 310021, China; 3Institute of Biotechnology, College of Agriculture and Biotechnology, Zhejiang University, Hangzhou 310058, China; 4Department of Plant Pathology and MOA Key Lab of Pest Monitoring and Green Management, College of Plant Protection, China Agricultural University, Beijing 100193, China

**Keywords:** bayberry, decline disease, humic acid, physical and chemical properties, microbial community, metabolomics

## Abstract

In recent years, bayberry decline disease has caused significant damage to the bayberry industry. In order to evaluate whether humic acid can be used to effectively control the disease, this research examined the nutritional growth and fruit quality of bayberry, soil physical and chemical properties, soil microbial community structure, and metabolites. Results indicated that the application of humic acid not only improved the vigor and fruit quality of diseased trees, but also increased the diversity of microbial communities in the rhizosphere soil. A great increase was observed in the relative abundance of bacterial genus *Mycobacterium* and *Crossiella*; fungal genus *Fusarium* and *Coniosporium*. In contrast, a significant decrease was observed in the relative abundance of bacterial genus *Acidothermus*, *Bryobacter*, *Acidibacter*, fungal genus of *Geminibasidium* and *Mycena*. Analysis of redundancies (RDA) for microbial communities and soil characteristics showed that the main four variables, including available nitrogen, phosphorus, potassium, and calcium, had a great effect on the composition of bacterial and fungal communities in bayberry rhizosphere soil at the genus level. The main four variables had a greater effect on bacterial communities than on fungal communities. In addition, ABC transporter, arginine and proline metabolism, galactose metabolism, and glutathione metabolism were significantly affected by humic acid, which changed the content of 81 metabolites including 58 significantly down-regulated metabolites such as isohexonic acid and carinitine, and 23 significantly up-regulated metabolites such as acidic acid, guaninosuccinate, lyxose, 2-monoolein, epicatechin, and pentonolactone. These metabolites also significantly correlated with rhizosphere soil microbiota at the phylum, order, and genus levels. In conclusion, the results demonstrated the role of humic acid on plant growth and fruit quality, as well as rhizosphere soil characteristics, microbiota, and secondary metabolites, which provides novel insights into the control of bayberry decline disease.

## 1. Introduction

*Myrica rubra* is an important fruit tree in southern China, with a planting area of about 334,000 hectares and an annual output of about 950,000 tons. It is also distributed in Japan, France, Kenya, the United States, and Brazil [1,2]. Bayberry fruit is rich in juice, and it is sour, sweet, and delicious. It not only contains carbohydrates, organic acids, proteins, and other substances, but is also rich in vitamin C and anthocyanin components. It has a high nutritional value and health care. Additionally, bayberry has been considered to be an important medicinal plant, indeed, its extract contains antioxidants that are effective against inflammation, allergies, diabetes, cancer, bacterial infection, and diarrhea [1,2].

However, in recent years, a major disease—the decline disease has occurred in the bayberry production area, mainly in the orchards during the peak production period. It is mainly manifested in poor tree growth, low fruit quality, unbalanced nutrient contents and trace elements deficient in the soil [3]. Therefore, it is very urgent for the industry to develop effective techniques for rejuvenating diseased trees. Interestingly, more and more attention has been paid on the role of humic acid (HA) in heathy plants. Indeed, HA is not only a component of humus, but also an organic macromolecule produced by the microbial biodegradation of plant, animal, and microbial residues. Due to the presence of several active functional groups (phenolic hydroxyl and carboxyl), HA has been regarded as a biocompatible, environmentally friendly, and affordable biosurfactant [4,5].

Previous studies have found that HA can improve the physical and chemical properties of soil [6,7], reduce the damage caused by abiotic stress, contribute to making plants more resistant [8], enhance the antioxidant system, promote making cell walls, and stabilize the metabolism of protein and sulfur-containing substances at the molecular level [9]. Furthermore, it can also promote plant growth, increase yield, and stimulate root absorption [10,11]. In addition, it can reduce the incidence rate of plant diseases [12,13]. In pre-experiment, we found that following the addition of HA to the soil, diseased bayberry trees can grow better, and their fruits taste better, reducing the number of cases of declining disease [14]. Therefore, it is very necessary to elucidate the mechanism of humic acid on the prevention and control of decline disease and plant growth promotion in bayberry.

The relationship between soil microorganisms and soil quality and fertility is reciprocal. It has been shown that soil metabolites participate in interactions between microbes and plants by regulating many growth processes of plants, development, and stress response processes [15]. Soil metabolites such as sugars, organic acids (amino acids and fatty acids), and secondary metabolites are crucial chemicals that mediate the ecological relationship between plants and other organisms. They can play a crucial role in plant growth, development, defense, and other physiological activities [16]. Several soil metabolites are harmful to plants. For example, esters can inhibit plants from respiration, degrade the structure of plant cells, and make it more difficult for bacteria to live in rhizosphere soil [17]. In contrast, some soil metabolites can influence the soil chemical and physical properties, regulate abiotic and biological processes, and control microbial communities, making them advantageous to plants [18].

For example, phenolic compounds are vital secondary metabolites that play a significant role in nutrient acquisition, allelopathy, pH regulation, and other ecological processes [19]. In addition to participating in the detoxification of plant reactive oxygen species, they suppress weed growth and can be utilized to build herbicides [20]. Increasing the amount of phenol and flavonoid metabolites in maize rhizosphere soil can control nitrogen fixation and assist beneficial bacteria to develop more [21]. Therefore, greater emphasis should be placed on the role of root metabolites in promoting plant growth and health.

This study investigated the impact of humic acid on vegetative growth, fruit quality, soil physical and chemical properties, soil microorganisms, and soil metabolites of bayberry. The role of humic acid in the prevention and control of decline disease offered a scientific foundation for developing the enhanced technology of the soil surrounding the root of bayberry and promoting the sustainable development of the bayberry industry.

## 2. Results

### 2.1. Effect of Humic Acid on Vegetative Growth and Fruit Quality

The findings of this study demonstrated that the vitality of weakened trees was considerably enhanced compared to the control after one year of application of humic acid. In fact, the treatment by humic acid resulted in increases of 28.62%, 11.03%, 12.13%, 14.86%, 10.68%, and 17.75% in the branch length, branch diameter, leaf length, leaf width, leaf thickness, and chlorophyll content, respectively, as compared to the control (Table 1). Similarly, the treatment by humic acid significantly increased fruit weight, soluble solids, total sugar, and vitamin C content by 36.30%, 16.19%, 24.72%, and 173.15%, but significantly decreased titratable acid levels by 50% (Table 2). This data revealed that humic acid had a mitigating effect on bayberry decline disease.

In agreement with the result of this study, our previous studies have shown that HA had the same effect on bayberry as fertilizer and fungicide on the growth of the plant and the quality of the fruit [3,22]. Furthermore, these data are also comparable to the findings of Nardi et al. [11], who found that humic acid can directly impact the photosynthesis of plants. In addition, the application of humic acid can significantly increase the total root length, root surface area, root tip number, and root volume of cucumber seedlings under low nitrogen conditions and promote plant height, stem diameter, and leaf area [23].

### 2.2. Effect of Humic Acid in Microbial Community Diversity

There was a difference in the number of operational taxonomical units (OTUs) between the presence and the absence of humic acid in the bacterial V3 + V4 region and fungal ITS region of the decline diseased bayberry trees. In fact, the average number of bacterial OTUs in the absence of and presence of humic acid are 1352.33 (range: 1202 to 1497) and 1564.83 (range: 1315 to 1764) (Figure 1), while the number of bacterial OTUs in the rhizosphere soil of decline diseased trees in the presence of humic acid is 15.17% higher than that of the absence of humic acid.

The average number of fungal OTUs in decline diseased bayberry trees was 750.67 (varying from 594 to 886) and 790.33 (varying from 761 to 845), respectively, in the absence and the presence of humic acid (Figure 1). In contrast, the number of fungi OTUs of rhizosphere soil in decline diseased bayberry trees was 5.28% higher in the presence of humic acid than in the absence of humic acid. Therefore, it suggests that humic acid had a greater effect on the bacterial operational taxonomic units than that of fungi of rhizosphere soil in decline diseased bayberry trees.

Compared with the decline diseased trees, the application of humic acid caused a 11.03% and 3.09%, respectively, increase in the Chao1 index and Shannon index of bacteria; a 2.76% and 4.74%, respectively, increase in the Chao1 index and Shannon index of fungi in the rhizosphere of bayberry (Figure 1). The result of this study suggests that humic acid had a greater effect on the bacterial richness of rhizosphere soil of declining diseased trees than on fungi. In agreement with the result of this study, our previous studies have shown that HA had the same effect on the variety and richness of the rhizosphere soil of decline diseased bayberry as a compound fertilizer and fungicide [22,24].

### 2.3. Effect of Humic Acid in Soil Microbial Community Structure

According to the principal coordinate’s analysis (PCoA) of the bacterial community structure, the six replicates of the diseased control and humic acid treatments were split into two groups. The diseased control was well separated from humic acid treatments, demonstrating that humic acid considerably affected the bacterial community structure of rhizosphere soil (Figure 2A). Furthermore, PCoA analysis of fungal community structure suggested that six replicates of the diseased control and humic acid treatment were divided into two distinct groups (Figure 2B). In addition, six replicates of the humic acid treatments revealed a better bacterial diversity compared to the diseased control than those of the fungal community structure (Figure 2). In accordance with the findings of this study, the structure and percentage contribution of soil microorganisms differed between fertilizer or fungicide-treated and untreated soil [22,24].

This finding demonstrated that the application of humic acid resulted in a significant change in the bacterial and fungal community structure at the phylum (Appendix A), order (Appendix A), and genus levels (Figure 3) compared to the decline disease control. Among the top 15 bacterial genera, *Acidothermus*, *Mycobacterium*, *Bryobacter*, *Acidibacter*, and *Crossiella* were the main five genera (Figure 3A). Compared with the decline disease control, humic acid treatment significantly increased the relative abundance of *Mycobacterium* and *Crossiella* by 16.06% and 25.32%, respectively; while it significantly reduced the relative abundance of *Acidothermus*, *Bryobacter*, and *Acidibacter* by 23.74%, 36.38% and 30.50%, respectively (Figure 3A). Interestingly, the beneficial effect of *Mycobacterium* and *Crossiella* on plant growth has been documented in previous studies [25,26]. For example, *Crossiella* exhibits bacteriostatic function [25], whereas *Mycobacterium* is helpful to plant growth. *Mycobacterium* Mya-zh01 can efficiently synthesize and secrete the plant growth hormone indole-3-acetic acid, resulting in the increase of the number and length of roots, plant height, number of leaves, and leaf length of *Doritaenopsis* [26].

Additionally, putting Mya-zh01 on *Doritaenopsis* seeds made them grow faster [26]. In vitro and in silico studies revealed that the strains prevented pathogenic Gram-positive and Gram-negative bacteria and fungi from growing. These results indicated that *Mycobacterium* and *Crossiella* may be advantageous for bayberry growth. The *Acidibacter* reduction is similar to the declined diseased bayberry treated by compound fertilizer and bio-organic fertilizer [3]. However, in contrast with the result of this study, *Bryobacter* has been found to contribute to high maize yield by increasing phosphorus flow and inhibiting toxic aluminum and manganese flow from soils to plants [27]. Among the top 15 fungi, *Cladophialophora*, *Geminibasidium*, *Mycena*, *Fusarium*, and *Coniosporium* were the main five genera (Figure 3B).

The relative abundance of *Fusarium* and *Coniosporium* increased significantly by 80.74% and 29.87%, respectively, compared with the decline diseased trees; the relative abundances of *Geminibasidium* and Mycena were significantly reduced by 30.71% and 86.27%, respectively, and no significant change was observed on *Cladophialophora* (Figure 3B). *Fusarium* spp. is a well-known soil-borne plant pathogen that causes severe vascular wilt in economically important crops worldwide [28]. The function of the *Coniosporium* has not been reported. *Geminibasidium* reduction in decline disease bayberry rhizosphere soil is comparable to a previous study in the rhizosphere soil [3] but different from compound fertilizer and bio-organic fertilizer treated soil [24]. By increasing nitrogen absorption and NH4+ assimilation, the plant growth-promoting fungus MF23 (*Mycena* sp.) increases the yield of *Dendrobium officinale* (Orchidaceae) [29]. Compared to the previous reports, *Fusarium* spp., *Geminibasidium* and *Mycena* may have distinct functions in bayberry rhizosphere soil.

In conclusion, beneficial bacteria increased while harmful bacteria decreased in bayberry root soil. Beneficial bacteria can utilize organic matter, degrade pollutants, improve stress tolerance, and enhance the growth of diseased bayberry trees to make them develop towards the level of healthy trees, reduce bayberry disease, and promote healthy bayberry tree growth.

### 2.4. Effect of Humic Acid on Soil Nutrient Status

Compared with the decline in diseased control, the content of pH, organic matter, alkali hydrolyzed nitrogen, exchangeable calcium, and exchangeable magnesium increased by 7.74%, 174.82%, 231.91%, 335.93%, and 316.10%, respectively, after applying humic acid treatment. In comparison, the content of available phosphorus and available potassium decreased by 86.48% and 72.96%, respectively (Table 3). Fourier-transform infrared (FTIR) spectroscopy revealed the complexation of Zn sulfate by HA through its S and C functional groups. In both oxisols, solution Zn increased due to the combined use of Zn and HA [30].

In the oxisol with lower organic matter content, HA can assure an adequate supply of residual Zn while increasing the growth of brachiaria cultivated in sequence with maize. Calcium phosphate particles coated with humic substances are a potential plant biostimulant from the circular economy. The obtained material showed promising results in its potential to elicit phosphorous uptake and foliar translocation by plants [31]. Humic acid reduces harmful aluminum ions in the soil, lowers soil acidity, promotes the release of alkali hydrolyzable nitrogen, exchangeable calcium, and exchangeable magnesium, and improves plant uptake, utilization, and transport of phosphorus and potassium. Improved bayberry soil physical, chemical, and nutritional balance.

### 2.5. Effect of Humic Acid on RDA of Soil Properties and Microbial Communities

The composition of bacterial and fungal communities in bayberry rhizosphere soil was affected significantly by different soil physical and chemical properties at the genus level (Figure 4) (Table 4). In the RDA result, the first and second axes explained 71.13% and 75.78% of the cumulative variance of the rhizosphere microbial community-factor correction, respectively, at the bacterial genus level (Figure 4A), 31.22% and 39.28% of the cumulative variance of the rhizosphere microbial community-factor correction, respectively, at the fungal genus level (Figure 4B). For the bacterial communities at the genus level, the contributions of four main variables were shown as below: 28.8% for exchangeable calcium, 25.0% for available phosphorus, 16.5% for available potassium, 13.0% for available nitrogen, and 10.0% for pH. The contribution of organic matter to bacteria (4.2%) was smaller than to fungi (32.6%), which may be because the decline disease was caused by a fungal pathogen, resulting in a greater number of fungi than bacteria in the rhizosphere of diseased bayberry trees [32].

The contributions of four main variables, including organic matter, available phosphorus, pH and available potassium, explained 32.6%, 18.5%, 14.7% and 9.4% of the fungal community, respectively, at the genus level. Results showed that the four main variables, including available nitrogen, phosphorus, potassium, and calcium, significantly affected the composition of bacterial and fungal communities in bayberry rhizosphere soil at the genus level. In agreement with the result of this study, a previous study has also shown that the growth of soil microbe is affected by a series of environmental factors such as soil pH, organic matter content, magnesium content, available nitrogen, phosphorus, and calcium [33]. However, this study revealed that the four main variables had a greater effect on bacterial communities than on fungal communities.

The result of this study revealed the complexity of the relationship between microbial growth and soil nutrient elements. Furthermore, the result obtained demonstrated that the order of contribution rates of soil physical and chemical indexes and nutrient elements to bacteria and fungi is different from that of untreated healthy trees, diseased trees, fertilizer-treated diseased trees and fungicide-treated diseased trees [3,22,24]. This data suggests that a balance of nutrients may be the best for microbial growth.

### 2.6. Change in Rhizosphere Soil Metabolomics

Using GC-MS analysis, 223 rhizosphere soil metabolites were identified in this study. Furthermore, a score map of metabolites was made using orthogonal partial least squares-discriminant analysis (OPLS-DA, a supervised statistical method of discriminant analysis), which can reduce intragroup differences, increase intergroup differences, and remove the impact of irrelevant factors on the experimental data to make accurate predictions of different samples. According to these findings, humic acid exhibited a great influence in the metabolomics distribution of rhizosphere soils from decline diseased bayberry. Indeed, in the absence of HA, the decline diseased samples were all distributed in the negative area of t [1] (principal component 1).

In contrast, in the presence of HA, the decline disease samples were all distributed in the positive area of t [1] (Figure 5). Furthermore, the parameters of the OPLS-DA are R^2^X(cum) = 0.574, R^2^Y(cum) = 0.997, Q^2^(cum) = 0.881. The Q^2^ values were greater than 0.5, which showed that the model had good interpretation and prediction ability. The cluster separation effect of LD and HA indicated that the application of humic acid significantly changed the metabolic structure of the rhizosphere soil of the decline-diseased bayberry.

### 2.7. Analysis of Differential Metabolites

The cluster analysis showed the contents and changes of metabolites in rhizosphere soil under LD and HA, while the major metabolites with significant changes between LD and HA were sugar, amino, organic, and secondary. Among the 81 significantly changed metabolites, 23 were up-regulated by 11.30%–3547.82%, with aconitic acid being the most, and 58 were down-regulated by 15.55%–79.49%, with deoxycholic acid being the least (Figure 6 and Table 5). The 19 metabolites with more than 50% reduction in relative contents were palatinitol, 1-monoolein, zymosterol, deoxycholic acid, hexitol, maltotriose, melezitose, isohexonic acid, carinitine, proline, 1-kestose, Tagatose and so on. However, the exogenous application of 1 mM carnitine mitigates the harmful effects of salt stress by increasing mitosis and decreasing DNA damage caused by oxidative stress on barley seedlings [34].

Proline is a multifunctional amino acid involved in plant adaptation to environmental constraints [35]. As the main fructooligosaccharide, 1-kestose shares similar physiological effects with other fructooligosaccharides. They have recently been determined to show more notable effects in promoting the growth of probiotics, including Faecalibacterium prausnitzii and Bifidobacterium, than those of other fructooligosaccharides [36]. Tagatose is a rare sugar decomposed by a few microorganisms and inhibits a broad range of phytopathogens [37]. In this study, the application of humic acid may increase the absorption and utilization of these substances by plants. Hence, the content of these secondary metabolites in the soil is lower than that of the control.

The 23 metabolites with increased relative content may be beneficial to plant growth. The 12 metabolites with >50% increase of relative levels increased were 5-methoxytryptamine, diclofenac, pentonolactone, epicatechin, 2-monoolein, lyxose, 1-hexadecanol, guanidinosuccinate, beta-sitosterol, coniferin, aconitic acid, and dehydroabietic acid. The six relative contents of aconitic acid, guanidinosuccinate, lyxose, 2-monoolein, epicatechin, and pentonolactone were increased by 1101.31%-3547.28%. Interestingly, diclofenac could potentially act as an environmental contaminant disturbing the natural developmental processes of plants [38]. Furthermore, beta-sitosterol has been regarded as an important plant growth regulator [39]. Therefore, the effect of humic acid on the growth of bayberry trees may be, at least partially, attributed to the increased release of the above metabolites in rhizosphere soil.

Compared to control cells, salt-adapted cells accumulated higher sugars, amino acids, and intermediate metabolites in the shikimate pathway, such as coniferin [40]. Aconitic acid plays various biological roles within cells as an intermediate in the tricarboxylic acid cycle. It confers unique survival advantages to some plants as an antifeedant, antifungal, and means of storing fixed pools of carbon [41]. Aconitic acid has also been reported as an inhibitor of fermentation, an anti-inflammatory, and a possible nematicide [41]. The time-kill assay with 6.25 g·ml^−1^ of dehydroabietic acid showed that it killed *Staphylococcus* epidermidis (American Type Culture Collection 14,990) bacteria within 24 h [42]. The application of humic acid may increase the release of beneficial substances from the soil, reduce the absorption of toxic substances by plants, and increase the absorption of beneficial substances.

### 2.8. Effect of Humic Acid on Metabolic Pathways

Using the KEGG (Kyoto Encyclopedia of Genes and Genomes) database, a pathway enrichment analysis of various metabolites in LD and HA was performed. There were significant differences between LD and HA in 10 metabolic pathways, including ABC transporters (*p* < 0.01), aminoacyl-tRNA biosynthesis (*p* < 0.01), galactose metabolism (*p* < 0.01), glutathione metabolism (*p* < 0.01), butanoate metabolism (*p* < 0.01), cyanoamino acid metabolism (*p* < 0.01), arginine and proline metabolism (*p* < 0.01), phosphotransferase system (PTS) (*p* < 0.05), amino sugar and nucleotide sugar metabolism (*p* < 0.05), and glycine, serine and threonine metabolism (*p* < 0.05) (Figure 7). The results demonstrated that the application of humic acid increased the concentration of metabolically active metabolites.

The amino acid metabolism, sugar metabolism, and nucleotide metabolism are the principal metabolic pathways of the differential metabolites in rhizosphere soil from diseased bayberry trees between the absence and presence of humic acid. Interestingly, ABC transporters not only have a detoxification mechanism, and play an important role in the response of plant pathogenic microorganisms, regulation of heavy metals, and secondary metabolite transport, but also affect the absorption of nutrients by bacteria [43]. Furthermore, glutathione metabolism has a detoxifying effect, and the increase of its metabolic level is conducive to plant growth and microbial reproduction.

Arginine and proline are important amino acids in plants, while arginine has the functions of ammonia detoxification, hormone secretion, immune regulation, and nitrogen storage. At the same time, proline is not only the energy which restores growth, but also has the function of maintaining protoplasm, environmental osmotic balance, and preventing water loss, and its metabolism also plays a good role in cold resistance, drought resistance and salt tolerance [44,45]. Therefore, the application of humic acid may improve the metabolic level of debilitated bayberry by promoting the transport of substances in the body and the absorption of nutrients by bacteria in the soil through ABC transporters.

Through galactose metabolism and glutathione metabolism, humic acid can inhibit harmful microorganisms and promote the growth of beneficial microorganisms, so as to improve the growth status of weak and diseased bayberry trees. Through the metabolism of arginine and proline, the resistance to cold stresses will be improved, which was able to promote the recovery and healthy growth of the root system of bayberry. The significant change in galactose metabolism after humic acid treatment is the same as those of diseased trees treated with fungicides and fertilizers, but the number of pathways was greater than that of fungicides and fertilizers [22,24].

### 2.9. Correlation of Soil Microorganisms with Metabolites

The relative content of bacteria at the phylum (Appendix A), order (Appendix A), and genus (Figure 8) levels are correlated with the relative content of the main secondary metabolites in the soil. The bacterial genera closely related to the content of secondary metabolites were *Jatrophihabitans*, *Bryobacter*, *Candidatus Soliactor*, and Rhiodiomicrobium, which were significantly related to 33, 23, 21, and 12 metabolites, respectively. In detail, *Jatrophihabitans* and *Rhizomacrobium* were negatively correlated with 19 and 12 metabolites, respectively, with reduced relative content and positively correlated with 14 and 0 metabolites, respectively, with increased relative content.

Furthermore, *Bryobacter* and *Candidatus Soliactor* were positively correlated with 17 and 13 metabolites, respectively, with reduced relative content and negatively correlated with 6 and 8 metabolites, respectively, with increased relative content. *Crossella* and *Mycobacterium* with significantly increased relative content were correlated with 1 and 0 of 58 metabolites, respectively. *Acidibacter* and *Acidothiermus* with significantly reduced relative content were significantly correlated with the contents of 5 and 4 secondary metabolites, respectively. This data shows that some functional bacteria play an important role in the change of the content of secondary metabolites.

The relative content of fungi at the phylum (Appendix A), order (Appendix A), and genus (Figure 9) levels are correlated with the relative content of the main secondary metabolites in the soil. The fungi closely related to the content of secondary metabolites are *Mortierella*, *Umbelopsis*, *Rickenella*, *Penicillium*, *Fusarium* and *Geminibasidium*, which are significantly related to 39, 38, 27, 15, 13, and 10 metabolites, respectively. There were 17, 7, and 3 species of *Mortierella*, *Penicillium*, and *Fusarium* that had a significant positive correlation with metabolites with increased relative content, and 22, 8, and 10 species of *Mortierella*, *Penicillium*, and *Fusarium* that had a significant negative correlation with metabolites with decreased relative content, respectively.

*Umbelopsis*, *Rickenella*, and *Geminibasidium* have 6, 4, and 0 species with significant negative correlation with metabolites with increased relative content, and 32, 23, and 10 species with significant positive correlation with metabolites with decreased relative content, respectively. This result shows that some functional fungi play an important role in the change of the content of secondary metabolites.

## 3. Materials and Methods

### 3.1. Experimental Design

This study was performed in July 2019 on 15-year-old Dongkui bayberry trees from Majian Town, Lanxi City. The tested bayberry trees with the similar load, crown size, leaf loss (10–25% of the whole plant’s leaves) and disease index (grade 3) [3]. The distance between planting rows was about 4 m × 5 m, while the orchard was managed routinely in the traditional manner. After harvesting the fruit, each decline diseased bayberry tree was treated by using 0 and 1400 g/plant of humic acid (90% effective content, Zhejiang Fengyu Ecological Technology Co., Ltd., Pujiang, China), which was applied in the drip line ditch of the roots. Each replicate consisted of a single tree, and each treatment contained 6 trees with similar vegetative and reproductive growth. The effect of humic acid on decline diseased bayberry trees was evaluated in June 2020 by measuring substantial variations in bayberry’s spring shoots, leaves, fruits, and the microbial community structure of rhizosphere soils.

### 3.2. Measurement of Vegetative Growth Parameters

The collection and measurement of branch and leaf samples were carried out, as described by Ren et al. [3]. In detail, the diameters of 20 randomly selected ripe spring branches and 30 randomly selected leaves from 6 trees in each treatment were measured with a ruler and a digital Vernier caliper (Shanghai Daoju, Shanghai, China). Leaves were sampled from the fourth to eighth leaves down from the top of the living branches in the middle of the tree’s circumference. The width and length (from tip to petiole) of the leaves were measured with a ruler, while the average thickness of leaves was determined with a digital Vernier caliper (Beaverton, OR, USA). The photosynthesis rate was calculated using a LI-6400 hand-held photosynthesis analyzer (LI-COR Company of America, Lincoln, NE, USA).

### 3.3. Measurement of Fruit Economic Characters

The collection and economic characters measurement of fruit samples were carried out, as described by Ren et al. [3]. In brief, 15 mature fruits were randomly selected from 6 trees in each treatment during fruit maturation phase. Fruit samples were routinely stored at −20 °C. After harvest, the weight and soluble solids content of single fruit were determined immediately. The fruit fresh weight was measured using an electronic balance (Shanghai Precision Instrument, Shanghai, China), while the contents of total soluble solids (TSS) was determined using an ATAGOPR-101a hand-held digital glucometer (Tokyo, Japan). In addition, the concentration of titratable acid in fruit samples was measured by acid–base titration, as described by [18], while the vitamin C concentration in fruit samples was quantified by using 2–6 dichloroindophenol titration, as described by Ghimire et al. [19].

### 3.4. Soil Sample Collection and Physical and Chemical Property Measurements

Soil sample collection and physical and chemical property measurements were performed, as described in our previous study [3]. After fruit harvest, rhizosphere soil samples were collected from the six trees in each treatment. A total of 2 kg of mixed soil samples (0–20 cm) were collected using the quartering method from the drip line surrounding the crown of the bayberry plant. After passing through a 0.45 mm filter, one-half sample was kept in a refrigerator at −80 °C for the extraction of genomic DNAs, and the other sample was air dried at room temperature to measure the soil physical and chemical properties. The content of organic matter was determined by using the K2CrO_7_ oxidation method, while the pH was measured using a pH meter with a soil-to-water ratio of 1:2.5 [21]. The available N and available P were evaluated with the alkaline hydrolysis diffusion method and the anti-molybdenum antimony colorimetry method, respectively [46]. The available K was determined with the ammonium acetate extraction flame photometer method [47]. After extraction with ammonium acetate and iron, the exchangeable calcium and magnesium were measured with an ice3500 atomic absorption spectrophotometer [20].

### 3.5. Soil Genome Sequencing

Sample processing and microbiome sequencing were performed according to the method of [3]. In brief, the DNAs were extracted from the rhizosphere soil samples by using the E.Z.N.ATM Mag-Bind Soil DNA Kit (Biodiz, Lince, Peru) following the manufacturer’s instructions (OMEGA, Norcross, GA, USA). The purity of the extracted DNAs was evaluated using a Nano-Drop (ND-1000) spectrophotometer (Thermo Fisher Scientific, Waltham, MA, USA). Bacterial diversity was determined by amplifying the V3–V4 region of the 16S rRNA gene using the primers 341F and 805R [48], while fungal diversity was determined by amplifying the ITS1 and ITS2 region using the primers ITS1F and ITS2 [3,15], respectively.

The components of the PCR included 1 μL DNA template, 1 μL (10 μM) each forward and reverse primer, 15 μL 2 × Hieff^®^ Robust PCR Master Mix (Yeasen, Shanghai, China), 12 μL ddH_2_O. Furthermore, the PCR amplicons were purified using Vazyme V AHTSTM DNA clean beads (Vazyme, Nanjing, China). Finally, equimolar ratios of the purified amplicons (Six in total) were pooled and subjected to 2 × 250 bp pair-end sequencing on the Illumina MiSeq system (Sangon Biotech (Shanghai) Co., Ltd., Shanghai, China).

### 3.6. Gas Chromatography-Mass Spectrometry (GC-MS) Metabolomics Analysis

For detailed methods of soil sample processing and secondary metabolite detection, we refer to our previously published paper [3]. In order to increase the accuracy of the entire analysis, we also included one quality control (QC) sample for every 10 samples. The QC sample was prepared by mixing the extracts of all samples in the same amount, and each QC sample was the same amount as the tested sample. Analysis of GC-MS metabolomics of soil samples was conducted on Agilent Technologies Inc. 7890B-5977A GC/MSD GC-MS (Santa Clara, CA, USA), which was carried out, as described in our recent publication [3]. Furthermore, the metabolite information was obtained by analyzing the KEGG database after comparing the data obtained in this study with the standard spectrum library.

### 3.7. Statistical Analysis

Preliminary data were processed using Excel 2016 statistical software with a significance level of 0.05 for *t*-test, while the collected data were compared with those in the standard spectrum library, which was established by the National Institute of Standards and Technology. Furthermore, PCoA, community histograms and redundancy discriminant analysis (RDA) were performed by using R 3.5.1 (The R Foundation for Statistical Computing, Vienna, Austria). In addition, the heat map was drawn by using R package, Pheatmap version 1.0.12 (https://CRAN.R-project.org/package=pheatmap, accessed on 15 October 2022). Orthogonal partial least squares discriminant analysis (OPLS-DA) was made by using R 3.6.2 ropls. The α-diversity metrics and carry out the significance test (*p* < 0.05) was calculated using the *t*-test and SPSS 17.0 software (IBM, Chicago, IL, USA).

## 4. Conclusions

The application of humic acid could improve the vigor of diseased trees, the fruit quality, and the diversity and species number of bacterial communities in the rhizosphere soil. A great increase was observed in the relative abundance of bacteria *Mycobacterium* and *Crossiella*, and fungi *Fusarium* and *Coniosporium*. In contrast, a great decrease was also found in the relative abundance of bacteria *Acidothermus*, *Bryobacter*, *Acidibacter*, and fungi *Geminibasidium* and *Mycena*. RDA of microbial communities and soil characteristics revealed that the composition of bacterial and fungal communities in bayberry rhizosphere soil was significantly affected by the main four variables, including available nitrogen, phosphorus, potassium, and calcium, at genus levels.

The four main variables exhibited a greater influence in bacterial communities than in fungal communities. In addition, the significant correlation between microorganisms and secondary metabolites at the genus level confirmed the alteration in the organization of the microbial community. According to the findings, there was a great difference between the absence and presence of humic acid in the metabolomics distribution of rhizosphere soils from decline disease bayberry. In fact, GC-MS metabolomics analysis showed that 81 metabolites in the rhizosphere soil of diseased trees changed significantly after applying humic acid, 23 metabolites such as acidic acid, guanidinosuccinate, lyxose, 2-monoolein, epicatechin, and pentonolactone were significantly up-regulated, and 58 metabolites such as 1-monoolein, zymosterol, deoxycholic acid, hexitol, maltotriose, isohexonic acid and carinitine were significantly down-regulated.

There was a great difference between the absence and presence of humic acid in 10 metabolic pathways, including ABC transporters, galactose metabolism, glutathione metabolism, aminoacyl-tRNA biosynthesis, butanoate metabolism, cyanoamino acid metabolism, arginine and proline metabolism, phosphotransferase system, amino sugar and nucleotide sugar metabolism, and glycine, serine, and threonine metabolism. Overall, this study highlighted the great influence of humic acid in decline disease by regulating soil microbial community, physical and chemical properties, and secondary metabolites in the bayberry rhizosphere, which provided a novel strategy for the management of bayberry decline disease.

## Figures and Tables

**Figure 1 ijms-23-14707-f001:**
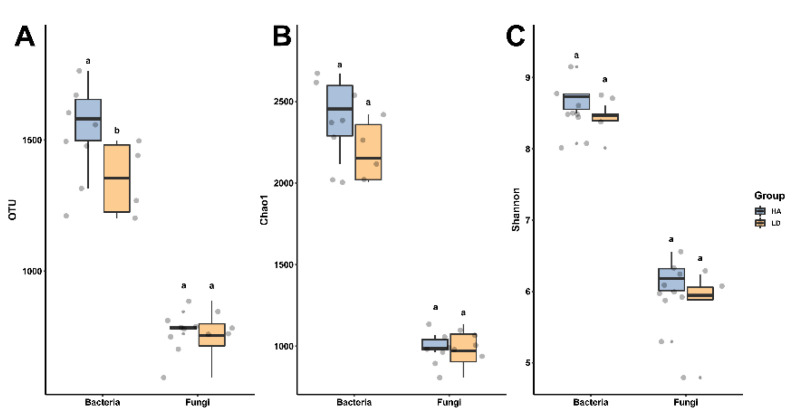
The influence of humic acid on the OTU (**A**), Chao1 (**B**) and Shannon (**C**). LD and HA represent decline diseased bayberry in the absence and presence of humic acid, respectively. Significant changes (*p* < 0.05) are indicated by values with distinct lowercase letters between the absence and presence of humic acid.

**Figure 2 ijms-23-14707-f002:**
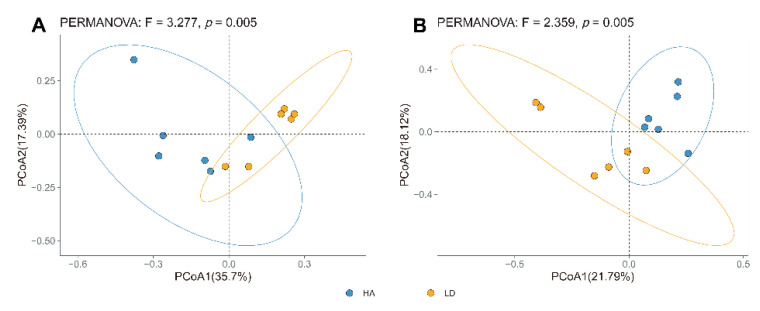
PCoA results (**A**) for soil bacteria and (**B**) for soil fungi based on OTU abundance. LD and HA represent the absence and presence of humic acid in diseased bayberry trees, respectively.

**Figure 3 ijms-23-14707-f003:**
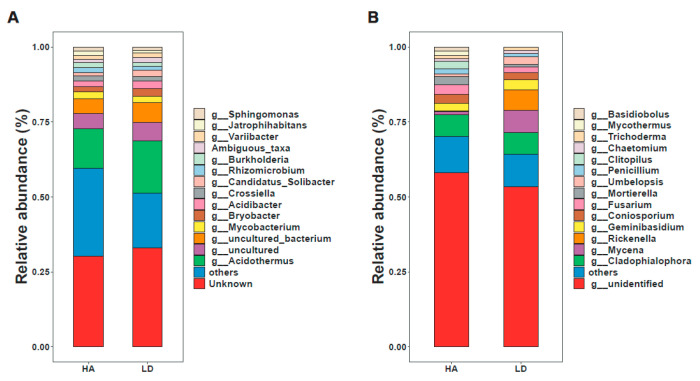
Relative abundance of bacteria (**A**) and fungi (**B**) at the genus level. LD and HA represent the absence and presence of humic acid in diseased bayberry trees, respectively.

**Figure 4 ijms-23-14707-f004:**
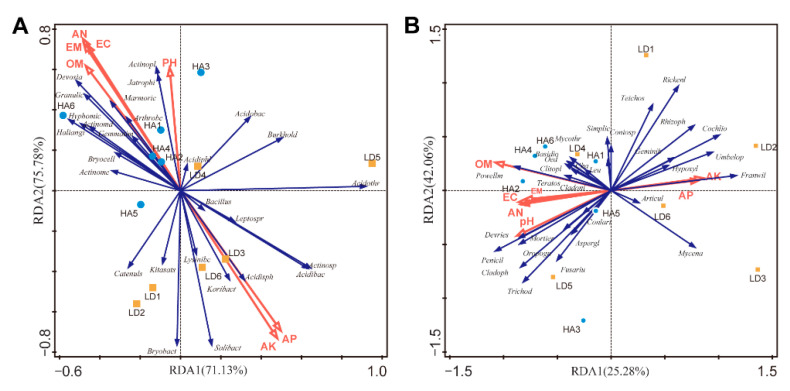
RDA of the rhizosphere microbial community composition at a genus level with soil physiological and chemical properties. RDA: Redundancy discriminant analysis; PH: pH; OM: organic matter; AN: alkali-hydrolyzable nitrogen; AP: available phosphorus; AK: Available potassium; EC: exchangeable calcium; EM: exchangeable magnesium; Yellow Square: samples from decline diseased bayberry; Blue Dot: sample from the humic acid treated trees; LD and HA represented the absence and presence of the humic acid in diseased bayberry trees. (**A**): Baceria. *Acidibac*: *Acidibacter*; *Acidiphl*: *Acidiphilium*; *Acidisph*: *Acidiisphaera*; *Acidobac*: *Acidobacterium*; *Acidothr*: *Acidothermus*; *Actinomd*: *Actinomadura*; *Actinomc*: *Actinomycetospora*; *Actinopl*: *Actinoplanes*; *Actinosp*: *Actinospica*; *Arthrobc*: *Arthrobacter*; *Bacillus*: *Bacillus*; *Bryobact*: *Bryobacter*; *Bryocell*: *Bryocella*; *Burkhold*: *Burkholderia*; *Koribact*: *Candidatus_Koribacter*; *Solibact*: *Candidatus_Solibacter*; *Catenuls*: *Catenulispora*; *Devosia*: *Devosia*; *Gemmatim*: *Gemmatimonas*; *Granulic*: *Granulicella*; *Haliangi*: *Haliangium*; *Hyphomic*: *Hyphomicrobium*; *Jatrophi*: *Jatrophihabitans*; *Kitasats*: *Kitasatospora*; *Leptospr*: *Leptospirillum*; *Lysinibc*: *Lysinibacillus*; *Marmoric*: *Marmoricola.* (**B**): Fungi. *Articul*: *Articulospora*; *Aspergl*: *Aspergillus*; *Basidio*: *Basidiobolus*; *Chaetom*: *Chaetomium*; *Cladoni*: *Cladonia*; *Cladoph*: *Cladophialophora*; *Clitopl*: *Clitopilus*; *Cochlio*: *Cochliobolus*; *Coniosp*: *Coniosporium*; *Conlari*: *Conlarium*; *Devries*: *Devriesia*; *Franwil*: *Franwilsia*; *Fusariu*: *Fusarium*; *Geminib*: *Geminibasidium*; *Hypoxyl*: *Hypoxylon*; *Leucocp*: *Leucocoprinus*; *Mortier*: *Mortierella*; *Mycena*: *Mycena*; *Mycothr*: *Mycothermus*; *Oedogon*: *Oedogoniomyces*; *Oropogn*: *Oropogon*; *Penicil*: *Penicillium*; *Powellm*: *Powellomyces*; *Rhizoph*: *Rhizophlyctis*; *Rickenl*: *Rickenella*; *Simplic*: *Simplicillium*; *Teichos*: *Teichospora*; *Teratos*: *Teratosphaericola*; *Trichod*: *Trichoderma*; *Umbelop*: *Umbelopsis*.

**Figure 5 ijms-23-14707-f005:**
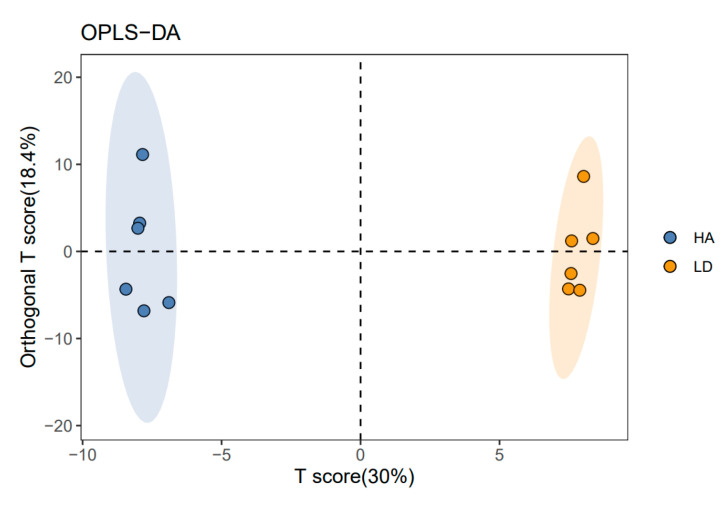
OPLS-DA score map of bayberry rhizosphere soil of the compound humic acid treatment. LD and HA represent the absence and presence of humic acid in diseased bayberry trees, respectively.

**Figure 6 ijms-23-14707-f006:**
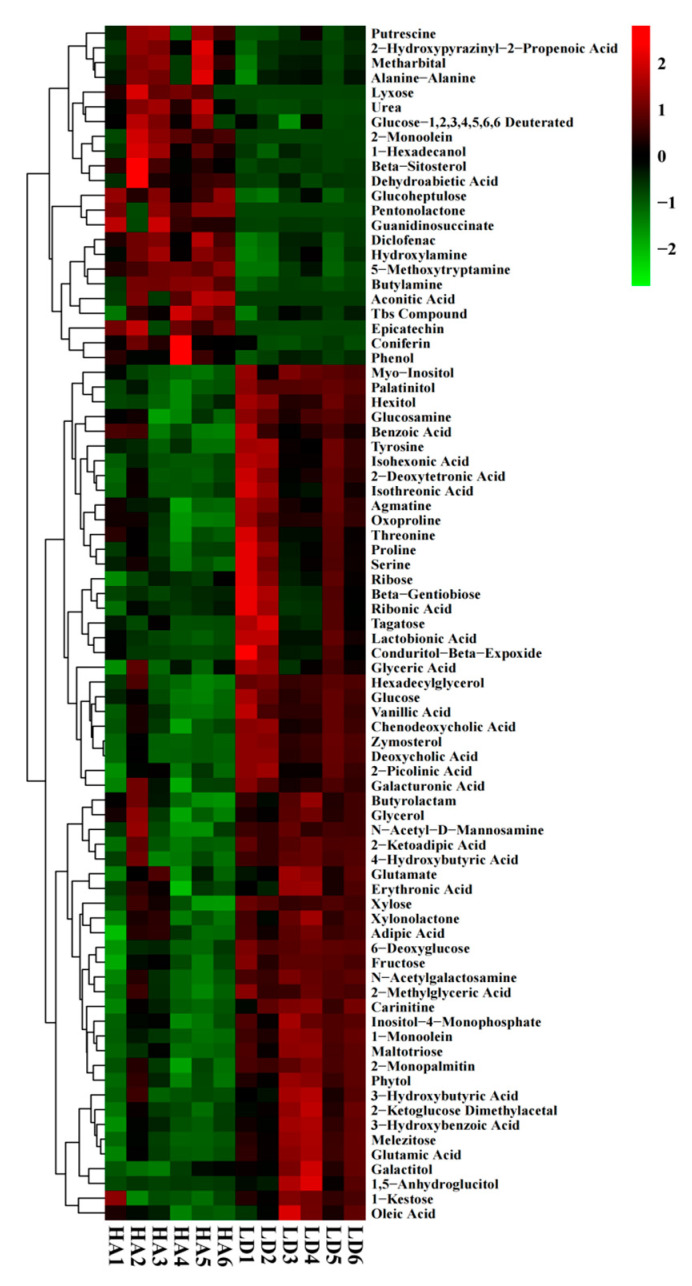
Thermogram analysis of different metabolites in bayberry rhizosphere soil. LD1-6 and HA1-6 represent the absence and presence of humic acid in diseased bayberry trees, respectively.

**Figure 7 ijms-23-14707-f007:**
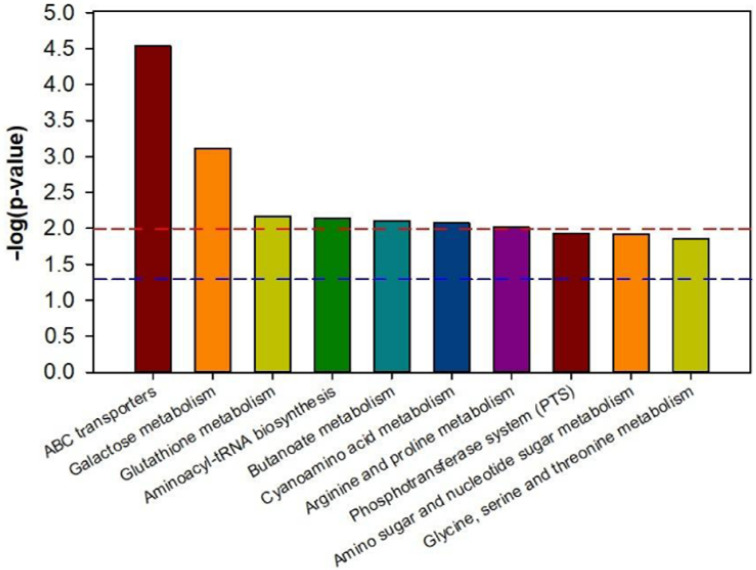
Metabolic pathway enrichment map of different metabolites in bayberry rhizosphere soil of the humic acid treatment. LD and HA represent the absence and presence of humic acid in diseased bayberry trees, respectively. The signal pathway is significant when the top of the bar is higher than the blue (*p* < 0.05) or red line (*p* < 0.01).

**Figure 8 ijms-23-14707-f008:**
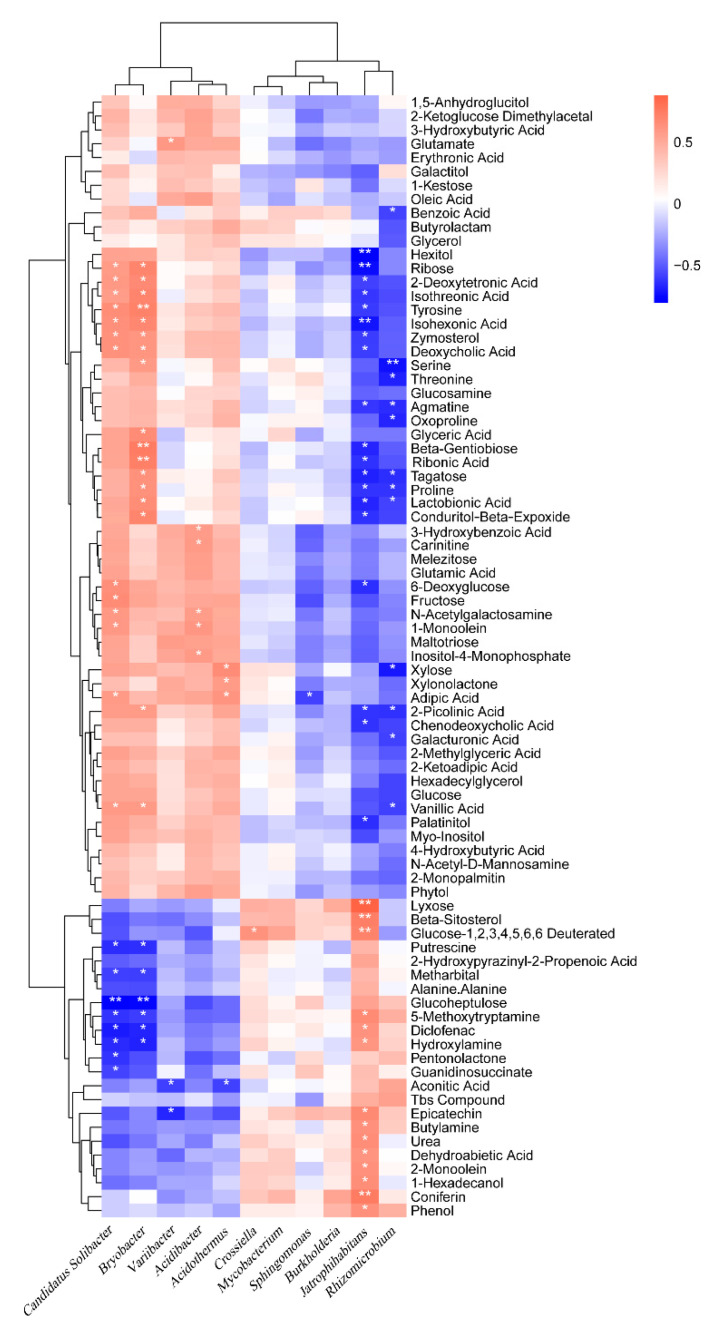
Correlation analysis between the microorganism relative abundances at the bacterial genus level and the relative metabolite contents of the humic acid treatment. “*” and “**” represents a significant correlation at *p* < 0.05 and *p* < 0.01, respectively. The magnitude of the correlation coefficient was indicated by the depth of the orange and blue scale, while the orange darker color is the greater positive correlation, and the blue darker color is the greater negative correlation.

**Figure 9 ijms-23-14707-f009:**
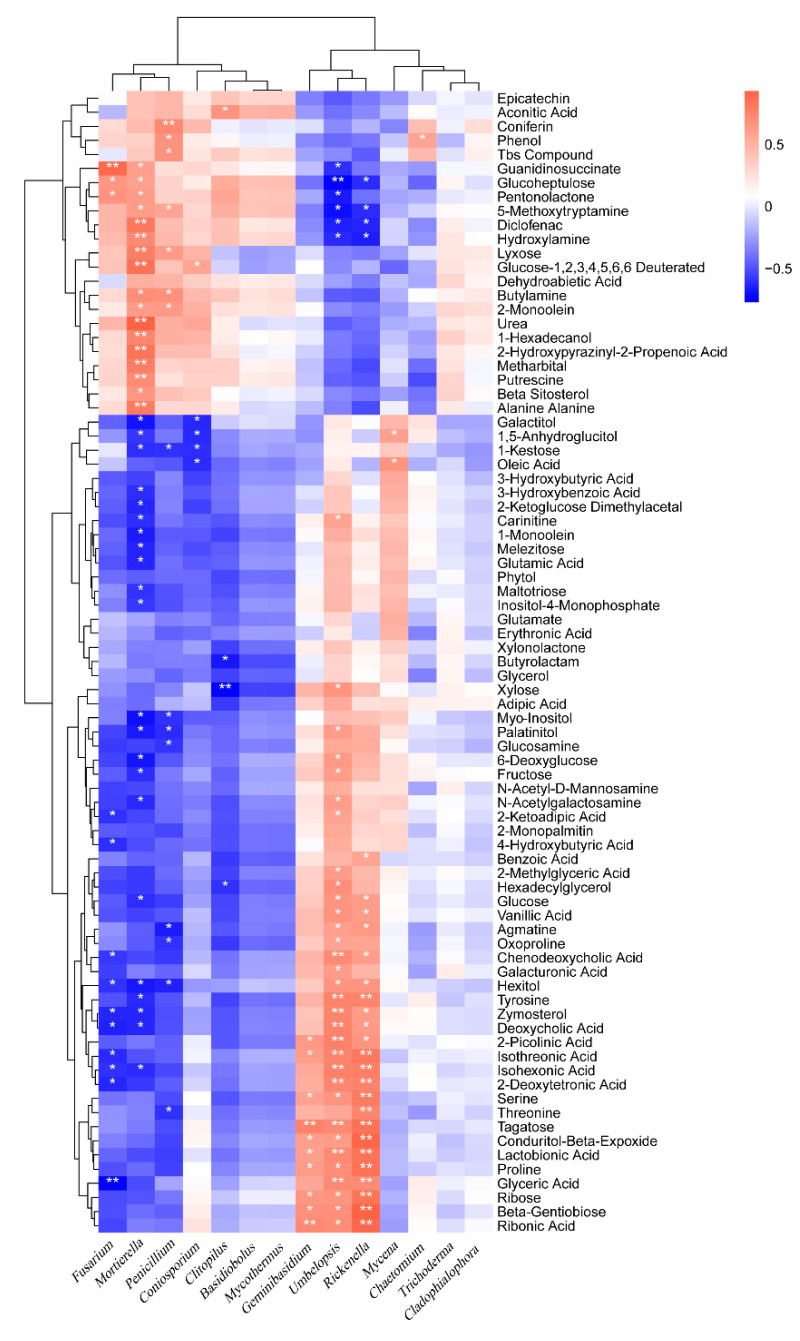
Correlation analysis between the microorganism relative abundances at the fungal genus level and the relative metabolite contents of the humic acid treatment. “*” and “**” represents a significant correlation at *p* < 0.05 and *p* < 0.01, respectively. The magnitude of the correlation coefficient was indicated by the depth of the orange and blue scale, while the orange darker color is the greater positive correlation, and the blue darker color is the greater negative correlation.

**Table 1 ijms-23-14707-t001:** Effects of humic acid on vegetative growth of decline diseased bayberry.

Parameters	Value	Parameters	Value
Branch length (cm)		Branch diameter (mm)	
LD	63.34 ± 4.76	LD	2.63 ± 0.09
HA	81.47 ± 0.93 *	HA	2.92 ± 0.03 *
Leaf length (mm)		Leaf width (mm)	
LD	107.79 ± 1.57	LD	32.84 ± 0.45
HA	120.87 ± 0.55 *	HA	37.72 ± 2.27 *
Leaf thickness (10 pieces/mm)		Chlorophyll/(SPAD)	
LD	4.68 ± 0.05	LD	41.06 ± 0.30
HA	5.18 ± 0.14 *	HA	48.35 ± 0.14 *

LD and HA represent decline diseased bayberry in the absence and presence of humic acid, respectively. “*” shows significant increases compared to the control of the decline diseased trees (*p* < 0.05).

**Table 2 ijms-23-14707-t002:** Effects of humic acid on fruit quality of decline diseased bayberry.

Parameters	Value	Parameters	Value
Single fruit weight (g)		Total soluble solids (%)	
LD	26.86 ± 0.74	LD	13.71 ± 0.69
HA	36.61 ± 0.13 *	HA	15.93 ± 0.20 *
Total sugar		Vitamin C (mg/100 g)	
LD	12.70 ± 0.03	LD	2.16 ± 0.00
HA	15.84 ± 0.78 *	HA	5.90 ± 0.06 *
Titratable acid (%)			
LD	0.04 ± 0.00		
HA	0.02 ± 0.00 #		

LD and HA represent decline diseased bayberry in the absence and presence of humic acid, respectively. “*” and “#” indicates significant increases or decreases compared to the control of the decline diseased trees (*p* < 0.05).

**Table 3 ijms-23-14707-t003:** Effect of compound humic acid on rhizosphere soil pH, physical and chemical properties of decline diseased bayberry.

Physical and Chemical Properties	LD	HA
pH	4.57 ± 0.27	4.92 * ± 0.34
Organic matter (%)	1.84 ± 0.34	2.37 * ± 0.13
Alkali hydrolyzed nitrogen (mg/kg)	45.71 ± 1.82	151.72 * ± 24.78
Available phosphorus (mg/kg)	205.70 ± 8.63	27.81 # ± 3.97
Available potassium (mg/kg)	432.20 ± 3.89	116.87 # ± 11.25
Exchangeable calcium (mg/kg)	137.32 ± 16.43	598.61 * ± 16.19
Exchangeable magnesium (mg/kg)	24.41 ± 2.36	101.55* ± 0.94

LD and HA represent the absence and presence of humic acid in diseased bayberry trees, respectively. “*” and “#” represents significant (*p* < 0.05) increase and decrease compared to the control of the decline diseased trees, respectively.

**Table 4 ijms-23-14707-t004:** Contribution of soil environment to bacteria and fungi taxa at the genus level.

Soil Environment	Contribution at the Bacterial Genus Level (%)	Contribution at the Fungal Genus Level (%)
pH	10.0	14.7
Organic matter	4.2	32.6
Available nitrogen	13.0	7.9
Available phosphorus	25.0	18.5
Available potassium	16.5	9.4
Exchangeable calcium	28.8	8.4
Exchangeable magnesium	2.6	8.4

**Table 5 ijms-23-14707-t005:** The relative contents of the metabolites in rhizosphere soil were changed by humic acid on the decline diseased bayberry trees.

Metabolite Name	Relative Content	Metabolite Name	Relative Content
Putrescine		2-Hydroxypyrazinyl-2-Propenoic Acid	
LD	223.48 ± 12.81	LD	124.87 ± 3.48
HA	260.99 * ± 32.39	HA	145.17 * ± 14.46
Metharbital		Alanine-Alanine	
LD	6.03 ± 0.47	LD	132.12 ± 6.12
HA	7.70 * ± 1.26	HA	147.06 * ± 13.59
Lyxose		Urea	
D	0.42 ± 0.09	LD	170.82 ± 1.40
HA	9.97 * ± 5.43	HA	204.50 * ± 16.49
Glucose-1,2,3,4,5,6,6 Deuterated		2-Monoolein	
LD	0.37 ± 0.03	LD	0.48 ± 0.06
HA	0.44 * ± 0.04	HA	7.94 * ± 4.22
1-Hexadecanol		Beta-Sitosterol	
LD	1.25 ± 0.16	LD	2.33 ± 0.08
HA	2.48 * ± 0.77	HA	4.07 * ± 1.16
Dehydroabietic Acid		Glucoheptulose	
LD	2.00 ± 0.21	LD	4.88 ± 0.37
HA	3.91 * ± 1.51	HA	6.49 * ± 0.50
Pentonolactone		Guanidinosuccinate	
LD	1.91 ± 0.03	LD	0.30 ± 0.13
HA	41.36 * ± 18.97	HA	3.58 * ± 2.17
Diclofenac		Hydroxylamine	
LD	4.61 ± 0.54	LD	22.48 ± 2.52
HA	7.05 * ± 0.79	HA	30.40 * ± 3.09
5-Methoxytryptamine		Butylamine	
LD	58.12 ± 10.05	LD	1.05 ± 0.02
HA	108.26 * ± 8.43	HA	1.29 * ± 0.10
Aconitic Acid		Tbs Compound	
LD	0.77 ± 0.11	LD	3.92 ± 0.18
HA	28.17 * ± 20.37	HA	4.44 * ± 0.46
Epicatechin		Coniferin	
LD	0.35 ± 0.05	LD	0.41 ± 0.09
HA	9.50 * ± 4.72	HA	0.70 * ± 0.19
Myo-Inositol		Palatinitol	
LD	51.82 ± 5.53	LD	12.43 ± 0.74
HA	27.57 # ± 5.15	HA	4.92 # ± 1.45
Hexitol		Glucosamine	
LD	4.61 ± 0.69	LD	1.02 ± 0.06
HA	1.76 # ± 0.55	HA	0.73 # ± 0.14
Benzoic Acid		Tyrosine	
LD	6.30 ± 0.94	LD	1.69 ± 0.32
HA	4.24 # ± 1.61	HA	0.86 # ± 0.19
Isohexonic Acid		2-Deoxytetronic acid	
LD	5.88 ± 1.28	LD	2.02 ± 0.28
HA	2.51 # ± 0.41	HA	1.36 # ± 0.19
Isothreonic Acid		Agmatine	
LD	2.82 ± 0.64	LD	1.26 ± 0.20
HA	1.75 # ± 0.32	HA	0.68 # ± 0.25
Oxoproline		Threonine	
LD	41.30 ± 4.88	LD	1.23 ± 0.29
HA	23.33 # ± 9.17	HA	0.80 # ± 0.24
Proline		Serine	
LD	5.97 ± 2.37	LD	1.28 ± 0.39
HA	2.27 # ± 1.06	HA	0.66 # ± 0.25
Ribose		Beta-Gentiobiose	
LD	246.68 ± 38.36	LD	0.97 ± 0.43
HA	192.67 # ± 20.11	HA	0.48 # ± 0.06
Ribonic Acid		Tagatose	
LD	8.23 ± 2.78	LD	47.61 ± 22.07
HA	5.30 # ± 0.87	HA	20.05 # ± 8.06
Lactobionic Acid		Conduritol-Beta-Expoxide	
LD	15.30 ± 4.80	LD	2.05 ± 1.15
HA	7.40 # ± 1.69	HA	0.63 # ± 0.28
Glyceric Acid		Hexadecylglycerol	
LD	6.83 ± 1.18	LD	2.68 ± 0.13
HA	5.05 # ± 1.27	HA	1.60 # ± 0.50
Glucose		Vanillic Acid	
LD	13.14 ± 1.23	LD	1.81 ± 0.15
HA	7.62 # ± 1.76	HA	1.27 # ± 0.18
Chenodeoxycholic Acid		Zymosterol	
LD	0.44 ± 0.06	LD	8.71 ± 1.22
HA	0.24 # ± 0.08	HA	2.11 # ± 1.46
Deoxycholic Acid		2-Picolinic acid	
LD	14.16 ± 2.26	LD	2.46 ± 0.35
HA	2.90 # ± 2.65	HA	1.42 # ± 0.41
Galacturonic Acid		Butyrolactam	
LD	1.13 ± 0.09	LD	3.09 ± 0.20
HA	0.74 # ± 0.27	HA	2.61 # ± 0.39
Glycerol		N-Acetyl-D-Mannosamine	
LD	160.69 ± 8.38	LD	0.90 ± 0.03
HA	132.88 # ± 26.61	HA	0.64 # ± 0.20
2-Ketoadipic Acid		4-Hydroxybutyric acid	
LD	3.53 ± 0.12	LD	8.08 ± 0.47
HA	2.32 # ± 0.60	HA	4.04 # ± 2.28
Glutamate		Erythronic Acid	
LD	0.22 ± 0.04	LD	0.29 ± 0.03
HA	0.14 # ± 0.05	HA	0.24 # ± 0.04
Xylose		Xylonolactone	
LD	0.04 ± 0.03	LD	0.11 ± 0.09
HA	0.80 # ± 0.15	HA	0.86 # ± 0.14
Adipic Acid		6-Deoxyglucose	
LD	1.04 ± 0.05	LD	13.40 ± 0.38
HA	0.76 # ± 0.19	HA	8.40 # ± 1.17
Fructose		N-Acetylgalactosamine	
LD	3.71 ± 0.15	LD	4.24 ± 0.23
HA	2.70 # ± 0.38	HA	2.21 # ± 0.77
2-Methylglyceric acid		Carinitine	
LD	1.00 ± 0.05	LD	5.33 ± 0.94
HA	0.69 # ± 0.13	HA	2.25 # ± 0.98
Inositol-4-Monophosphate		1-Monoolein	
LD	2.13 ± 0.24	LD	1.91 ± 0.25
HA	1.18 # ± 0.31	HA	0.69 # ± 0.22
Maltotriose		2-Monopalmitin	
LD	2.20 ± 0.31	LD	2.05 ± 0.06
HA	0.88 # ± 0.32	HA	1.50 # ± 0.22
Phytol		3-Hydroxybenzoic acid	
LD	0.92 ± 0.13	LD	8.23 ± 1.48
HA	0.51 # ± 0.17	HA	4.53 # ± 1.11
2-Ketoglucose Dimethylacetal		3-Hydroxybutyric acid	
LD	4.44 ± 1.15	LD	30.44 ± 9.12
HA	1.99 # ± 0.78	HA	13.47 # ± 7.17
Melezitose		Glutamic Acid	
LD	1.14 ± 0.22	LD	1.76 ± 0.24
HA	0.42 # ± 0.15	HA	0.93 # ± 0.21
Galactitol		1,5-Anhydroglucitol	
LD	21.08 ± 3.78	LD	6.71 ± 3.06
HA	12.55 # ± 2.63	HA	2.11 # ± 0.23
1-Kestose		Oleic Acid	
LD	1.14 ± 0.22	LD	0.77 ± 0.18
HA	366.82 # ± 33.77	HA	0.51 # ± 0.12
Phenol			
LD	23.03 ± 0.64		
HA	26.95 * ± 3.18		

LD and HA represent the absence and presence of humic acid in diseased bayberry trees, respectively. “*” and “#” represents significant (*p* < 0.05) increase and decrease compared to the control of the decline diseased trees, respectively.

## Data Availability

Access to the citations for these Sequence Read Archive metadata: PRJNA890045 at https://submit.ncbi.nlm.nih.gov/subs/ (accessed on 13 October 2022).

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
