# Peer review of "Effect of Humic Acid on Soil Physical and Chemical Properties, Microbial Community Structure, and Metabolites of Decline Diseased Bayberry"

_ijms, 2022, doi:10.3390/ijms232314707_

Round 1

Reviewer 1 Report

line 70: ..fruits taste...?

line 72: ..prevention...?

line 76: ...growth processes of plants...?

line 84: ..soil...?

line 104, 108: ..by...?

line 197-8, 200-204, 209,211, 212, 214,217, 219, 221, 223,225-6, 230-32: ..maybe italics ?...

line 274: Table 4 ?...

line 299: ...the best...

line 399: ..maybe the Italics... ?

line 414. What does it mean the number 1-6  (HA1..., LD1...)?

line 469-71, 475, 478, 480: ..maybe the Italics... ?

line 496: ..are..?

line 527: ...there were applied...

line 624, 626: The significant/great increase...?

line 625, 627-8:  ..maybe the Italics... ?

Reviewer 2 Report

I have read and analyzed with some effort the manuscript "Effect of Humic Acid on Soil Physical and Chemical Properties, Microbial Community Structure, and Metabolites of Decline Diseased Bayberry".

The topic discussed, the effect of humus on plants, has been a long-known topic for some time. Plants evolved in the presence of humus and with only humus as food. Only with the advent of agriculture and, more in particular in recent years, have we learned to grow plants in the presence of chemical fertilizers and with ever smaller quantities of organic matter in the soil. Now we are realizing that we have progressively destroyed our soils which, having lower and lower quantities of organic matter (the soil considered in this research contains 1.84%, when it is advisable not to go below 3%) are no longer able to retain nutrients, to maintain a rich, resilient and efficient microbial community, and therefore plants are increasingly exposed to disease and pathogen attacks.

The results reported in the manuscript are perfectly in line with what could have been expected and it is hoped that these results will stimulate farmers, to more careful agriculture to maintain a good amount of organic matter, and an abundant, biodiversified, and resilient microbial community, to favor its transformation into humus.

As I wrote at the beginning of my evaluation of the manuscript, I found it difficult to read it because it is written in English which must be carefully revised, especially syntactically and in sentence construction. In particular, the materials and methods are very lacking in information. Some suggestions are given below.

Regarding the research approach, however, many well-planned and well-conducted analyzes are reported, but their discussion is lacking and needs to be reviewed. In many cases it also needs to be simplified and repetitions must be removed.

In particular:

Line 25: RDA, for the first time enter "Redundance Analysis" in full.

Line 39: Metabonomics -> Metabolomics.

Line 54: “possible excess of available phosphorus”. By restoring a healthy microbial community and reintroducing mycorrhizae - which have now disappeared from agricultural soils due to the antifungal treatments carried out - there would no longer be a need for phosphate fertilizers because mycorrhiza could supply phosphorus to the plant in a controlled manner.

Line 104: “treatment of humic acid”. I struggled to find in the manuscript how long ago the treatment was done.

Table 1: better to use “branch” than “twig”. Also because "branches" are used in materials and methods.

Figure 1: I was unable to identify the "lowercase letters within the same treatment".

Paragraph 2.3. Effect of humic acid in soil microbial community structure. I did not quite understand the first sentence. PCoA has split into two groups HA and LD, but then you say that "in contrast desease control was well separated from humic acid treatment". Why you say in contrast?

Lines 203-207: Unless it is specifically demonstrated that Crossiella and Mycobacterium have a positive effect on the treated plants, you can only speculate that tis have happened in this case given their increase.

Lines 290-293: The above is quite obvious.

Line 295: It cannot be said that "the results have found", but, that "the results obtained demonstrate....".

Table 4. How do you explain that organic matter has so little effect on bacteria than fungi?

Line 334: "soil metabolomics were identified" -> soil metabolites were identified.

Line 383: What was the origin of the humus you used if diclofenac and other molecules used as anti-inflammatories and analgesics were present? From table 5 it appears to be present to a lesser extent also in untreated soil.

Lines 434-436: Remove this sentence, it is not relevant to the subject.

Figure 7 caption. What is prochloraz treatment?

3.1 Experimental Design: completely rewrite. But also the other paragraphs must be revised and partially rewritten.

Line 575: K2CrO7-> K2CrO7
